# Aseptic Rearing and Infection with Gut Bacteria Improve the Fitness of Transgenic Diamondback Moth, *Plutella xylostella*

**DOI:** 10.3390/insects10040089

**Published:** 2019-03-28

**Authors:** Jasmine Somerville, Liqin Zhou, Ben Raymond

**Affiliations:** Centre for Ecology and Conservation, Penryn campus, College of Life and Environmental Science, University of Exeter, Cornwall TR10 9FE, UK; jasmine.somerville@talktalk.net (J.S.); zhou.liqin@oamiccn.com (L.Z.)

**Keywords:** genetically modified insects, symbiosis, microbiome, transgenic, self-limiting, insect rearing, mutualism

## Abstract

Mass insect rearing can have a range of applications, for example in biological control of pests. The competitive fitness of released insects is extremely important in a number of applications. Here, we investigated how to improve the fitness of a transgenic diamondback moth, which has shown variation in mating ability when reared in different insectaries. Specifically we tested whether infection with a gut bacteria, *Enterobacter cloacae*, and aseptic rearing of larvae could improve insect growth and male performance. All larvae were readily infected with *E. cloacae*. Under aseptic rearing, pupal weights were reduced and there was a marginal reduction in larval survival. However, aseptic rearing substantially improved the fitness of transgenic males. In addition, under aseptic rearing, inoculation with *E. cloacae* increased pupal weights and male fitness, increasing the proportion of transgenic progeny from 20% to 30% relative to uninfected insects. Aseptic conditions may improve the fitness of transgenic males by excluding microbial contaminants, while symbiont inoculation could further improve fitness by providing additional protection against infection, or by normalizing insect physiology. The simple innovation of incorporating antibiotic into diet, and inoculating insects with symbiotic bacteria that are resistant to that antibiotic, could provide a readily transferable tool for other insect rearing systems.

## 1. Introduction

The environmental impact of chemical insecticides, and widespread resistance to these products, has meant that biological approaches to pest management are more important than ever [1,2]. Many biological control approaches rely on efficient rearing of insect pests or natural enemies. This is especially true of sterile insect technique (SIT) and inundative biological control methods that rely on mass production and release of populations that are unable to support themselves in the environment [3,4,5]. In addition, some biological control pathogens, such as baculoviruses, can only be produced *in vivo* [6]. There is also increasing interest in the mass production and rearing of insects for food and feed, based on the efficiency with which insects can produce fat and protein from low-quality diets [7]. In all these entomological applications, ensuring the health and fitness of insects is vital, and this is arguably particularly important for biological control programs where artificially reared insects must compete with wild insects for access to mates [8].

In the sterile insect technique, for example, mass-produced male pest insects are irradiated with gamma rays, leading to chromosomal irregularities at meiosis and the inability to produce healthy gametes [4]. Mass release of these sterile males ensures that mating occurs with wild females, who are unable to produce viable offspring [9]. This technique was first developed to eradicate New World screw-worm (*Cochliomyia hominivorax*), a highly damaging cattle parasite [4]. SIT eradicated the screw-worm from the United States and Central America and has been used to suppress or eradicate other pest insects, such as the Mediterranean fruit fly [5]. Although successful, SIT has some limitations. Large insect production facilities can be the source of accidental releases of wild-type insects, so insect production factories are best sited far from eradication zones. The fitness of the irradiated males can be very low, necessitating the release of very large numbers of males, while not all insect species can tolerate high levels of irradiation. Mass release in itself imposes technical challenges, including the difficulty of sex-separating males from females [5].

An analogous approach to the SIT addresses some of these issues. Genetically engineered insects carrying transgenes under the control of the ‘Tet-off’ genetic switch [10,11], can be configured to regulate conditional sex-specific expression of a given gene such that females will not survive beyond early instars [12,13]. Since these transgenes are dominant lethals, they impose strong fitness costs, meaning these transgenes are rapidly lost from populations after release. These transgenes are therefore ‘self-limiting’, in the sense that these costs drive a rapid decline in transgene frequency, post-release. Female-specific self-limiting transgenes ensure sex-separation by the elimination of the females, while supplementing larval feed with tetracycline (or suitable analogues) represses the lethal phenotype, allowing rearing in the laboratory [11]. Self-limiting transgenic males are typically marked with fluorescent protein genes [14], making it straightforward to monitor of self-limiting genes in experimental or wild populations. Field tests on the mosquito *Aedes aegypti* have shown that this self-limiting transgene technology can locally reduce populations by 95% [15,16,17].

Although self-limiting transgenic technologies have advantages over SIT [18], transgenes still impose fitness costs: Male fitness can be reduced relative to that of wild type insects, potentially because of insertional effects or low rates of transcription of transgenic elements in males [19] or because of adaptation to laboratory conditions, a problem for Lepidoptera in particular [20]. Critically for this study, we have previously observed variation in the fitness of self-limiting diamondback moth (DBM, *Plutella xylostella*) between laboratories, even when using standard insect stocks reared on carefully developed protocols that can yield strong-performing males [13,21]. Variability in insect quality with the self-limiting DBM, when moved off the tetracycline diet, may be related to the female larvae that die as young instars within rearing containers: these cadavers may provide a source of microbial contamination on the larval feed. Here, we aimed to test whether reducing the opportunity for contamination would improve adult fitness by using an aseptic rearing capable of producing axenic or gnotobiotic insects [22]. 

Diet can be an important factor in determining the fitness of laboratory-reared insects [23]. Many insects in laboratory studies are typically reared on a combination of antibiotics, commonly tetracycline and streptomycin [11]. Although tetracycline is necessary to suppress dominant lethal transgenes [11], antibiotics will further reduce the microbial diversity within insects [24,25]. In addition to the use of antimicrobials, the consumption of artificial diet *per se* can reduce microbial diversity within insects [24,26]. Reduced microbial diversity may increase the vulnerability of insects to some pathogens [27,28], while insect gut symbionts have a range of potentially beneficial roles that could impact male fitness. For instance, gut microbes may improve nutrient assimilation [29,30], aid in the production of mating pheromones [31], or play a general role in nutrition and regulation of host metabolism [32], although their role in mate preference is somewhat contentious [33].

Here, we aimed to test two hypotheses using the self-limiting DBM as a model insect. First, whether aseptic rearing is able to prevent potentially compromised insect fitness by reducing opportunities for contamination of artificial diet, and second, whether the addition of microbial gut symbionts can further increase the fitness of these transgenic insects reared in the laboratory. We selected *Enterobacter cloacae* as our focal gut symbiont as this species can form persistent associations with the Lepidopteran gut [34] and because *Enterobacter* spp. are a common component of the gut microbiota in a variety of insects including DBM [30,34,35]. Following from previous work, we used a population of DBM carrying a female-specific self-limiting gene developed by Oxitec Ltd. (Abingdon, UK), a strain carrying a fluorescent marker that allows efficient calculation of mating success [13].

## 2. Materials and Methods

Development of the self-limiting DBM (OX4319L, Oxitec Ltd., Abingdon, UK) has been described previously [13]. In brief, the self-limiting system was implemented using tetracycline controlled transactivator sequences [13]. Sex-alternate splicing of the *doublesex* (*dsx*) sequence allows the development of a female-specific lethal genetic system that is repressible by the provision of tetracycline, or suitable analogues, in the larval feed [13]. The self-limiting (SL) strain was constructed in the Vero Beach genetic background [13]. Wildtype (WT) insects in this study were the population VLSS, a stock produced by out-crossing Vero Beach with the diet-adapted NO-QA population, as described previously [21]. The gut microbe *Enterobacter cloacae* (isolate JJBC), forms a persistent association with the gut of Lepidoptera and was recovered from larvae of DBM feeding on Chinese cabbage, *Brassica pekinensis*, in the insectary in the Department of Zoology, University of Oxford. The *E. cloacae* strain used in these experiments was JJBC 11.1B Strep^R^, which is a spontaneous mutant that is able to grow in the presence of streptomycin. This enabled us to combine the use of antibiotic with the addition of a symbiont and therefore specifically test the benefit of symbiont inoculation in addition to aseptic rearing.

Standard rearing conditions followed published protocols [36] with some minor modifications. In brief, insects in standard conditions were reared in non-sterile 100 mm plastic tubs with a depth of 45 mm. Diet (F9221B, Frontier Agricultural Sciences) was autoclaved prior to pouring; vitamins (Vanderzant’s 4 g/L, Ascorbic Acid 3.6 4 g/L) were added after diet had reached a temperature of ≈ 60 °C. All *P. xylostella* eggs were surface-sterilised (2% sodium hypochlorite solution, and three washes of autoclaved water) and counted before being added to rearing containers.

Aseptic rearing conditions were adapted from protocols for culturing axenic and gnotobiotic *P. xylostella*, which uses sterile diet and the offspring of parents reared on antibiotics [22], with some modifications. Here, insects were reared in 90 mm Petri dishes on artificial diet (as above). Filter-sterilized vitamins (concentrations as above) and antibiotic (streptomycin 0.125 g/L) were added to diet before pouring 20 mL into each dish in a class 2 microbiological safety cabinet. All manipulation of diet and Petri dishes took place inside a microbiological safety cabinet. In both rearing treatments, insects were reared on quarter sections of diet and fresh diet added as older diet became dehydrated or consumed. Insect egg densities were controlled to ≈ 120 SL eggs per container and were surface-sterilised as above.

For both aseptic and standard conditions, three treatment groups were established; SL without *E. cloacae* (SL−Ec) and SL with *E. cloacae* (SL+Ec); while wild type insects without *E. cloacae* (WT−Ec) were included as a positive control and to assess fitness costs of transgenes relative to a standard outbred stock. Details of inoculation of insects with *E. cloacae* gut bacteria are given below. Treatment groups were incubated at 24 ± 1 °C. At the fourth instar, males and females from the WT−Ec treatment groups were separated and replaced on fresh diet; pupal weights were recorded for males only. Mean pupal weights were recorded for each rearing container, to avoid pseudo-replication. Surviving SL pupae from all treatment groups in both standard and aseptic conditions were counted and compared to initial egg counts.

Emerging neonate larvae, originating from surface sterilized eggs, were inoculated with *E. cloacae* (*Ec*) using 1000-fold dilutions of overnight culture (5 mL L-Broth with 50 µg/ml streptomycin, 30 °C, 150 rpm). Bacteria were diluted in sterile saline (0.85% w/v NaCl). Insects were inoculated by coating quarter sections of diet with diluted *Ec* sufficient to cover the diet surface (600 µL in standard conditions; 400 µL in aseptic conditions); diet was dried in a class 2 microbiological safety cabinet, before adding surface sterilized eggs on Parafilm strips just before egg hatch. In order to enumerate bacterial infections, fourth instar larvae (1–2 per rearing container) were homogenized in 135 μL saline, serially diluted, and 15 μL of the homogenate was plated out using a dilution range of 10^−1^ to 10^−4^ onto 2% LB agar.

Mating competition experiments took place in 30 × 15 × 15 cm mating cages using a release ratio of 10 SL males to each WT−Ec male and a total of 10 WT males and 10 females in each cage. Bacterial treatments, rearing with *E. cloacae* (+Ec) or without *E. cloacae* (−Ec), were replicated five times in the aseptic regime and four times in the standard rearing conditions.

Mating occurred over a three-day period at 24 ± 1 °C once pupae had emerged as moths. Eggs collected over this mating period were reared on standard conditions, as above. Again, when larvae had reached fourth instar, females were removed from all diet tubs. As the self-limiting construct contains a dominant heritable, fluorescent DsRed2 protein marker [13], pupae were sorted using a binocular microscope with Nightsea™ (Nightsea, Lexington, KY, USA) light source (excitation 510–540 nm) and 600-nm filter, enabling us to score the proportion of WT and SL male progeny for each cage.

Statistical analysis was carried out in R v3.4.3 (http://www.r-project.com) primarily using analysis of variance, and generalized linear models (GLMs). To test for differences between specific factor levels we used post-hoc treatment contrasts (*t*-tests) in model summaries or model simplification to test for significant loss of explanatory power after pooling treatment levels. Bacterial count data, which included many zero data points were analysed with Kruskal-Wallis non-parametric one-way ANOVA. Proportional data (survival, proportion of transgenic progeny) were analysed in GLMs with quasi-binomial error distributions to compensate for over-dispersion. All model assumptions were checked with graphical analysis of error distribution assumptions (qqplots, and plots of residuals versus fitted values/leverage). Raw data are provided in the Appendix A.

## 3. Results

### 3.1. Bacteria

To test for the efficacy of our bacterial inoculation, and to investigate the presence or absence of contaminating bacteria in experimental treatments, whole insect larvae were homogenized and plated out to test for the presence of culturable bacteria. Inoculation with gut bacteria led to the effective colonization of insects, while uninoculated insects lacked any culturable gut microbes (Kruskal-Wallis tests—aseptic rearing, χ^2^ = 295, df = 2, *p* < 0.0001; standard rearing χ^2^ = 43.1, df = 2, *p* < 0.0001; Figure 1). Out of 96 larvae sampled in the SL+Ec treatment group, only two did not contain *E. cloacae*, while one larva contained another bacterial morphotype. In standard conditions, all sampled larvae (*n* = 8) showed 100% inoculation, and there were clear differences in bacterial densities between inoculated and uninoculated insects (Figure 1).

### 3.2. Development and Weight Gain

Survival from the egg to pupae stage was recorded for all self-limiting genotypes. Here, there was a marginally non-significant trend with slightly elevated survival in the standard rearing conditions in comparison to aseptic rearing (GLM: F_1,77_ = 3.73, *p* = 0.057; Figure 2). Inoculation with *Ec* did not have a consistent effect on larval survival (GLM: F_1,76_ = 3.37, *p* = 0.070; Figure 2) and these treatments did not interact (F_1,75_ = 0.047, *p* = 0.83). Note that because the self-limiting gene was not repressed under these conditions, all female larvae should die, meaning that survival, on average, was not expected to exceed 50%.

To test the hypothesis that *E.cloacae* would affect SL male growth, pupal weights were scored for inoculated SL insect (SL+Ec), uninoculated SL insects (SL−Ec), and uninoculated wildtype insects (WT−Ec) in aseptic and standard conditions. Insect genotype and inoculation treatment significantly affected pupal weight, with WT pupae having the greatest weight in both rearing regimes (GLM: F_2,83_ = 80.6, *p* < 0.0001). Rearing regime also affected weight; on average insects reared in standard conditions produced heavier pupae (GLM: F_1,82_ = 36.4, *p* < 0.0001; Figure 3). The effect of *Ec* inoculation, however, depended on rearing regime (GLM: inoculation × rearing interaction F_2,80_ = 3.51, *p* < 0.03; Figure 3). Under aseptic conditions, the gut bacteria marginally increased weight (post-hoc contrast, *t* = 2.34, *p* = 0.0217), while under standard conditions the gut bacteria appeared to be slightly parasitic and decreased pupal weight (post-hoc contrast, *t* = 2.28, *p =* 0.025). Overall, we confirmed a significant impact of *Ec* inoculation through model simplification; collapsing the *Ec* treatment groups for the self-limiting insects had a significant effect on pupal weight (GLM: F_2,82_ = 3.38, *p* < 0.038).

### 3.3. Fitness in Mate Competition Experiments

Here, the hypotheses that rearing conditions and *E. cloacae* inoculation would affect SL male mating success was tested by comparing the proportion of male progeny sired by transgenic and WT males. The presence of a semi-dominant GFP fluorescent marker in OX4319L males means that heterozygous WT/SL progeny can be readily identified, allowing us to make accurate measures of mate competitiveness [13]. The strongest effect was the increased fitness of SL males in the aseptic rearing regime versus those reared in standard conditions (GLM: F_1,14_ = 105, *p* < 0.0001; Figure 4). However, inoculation with *Ec* also improved the ability of males to compete for mates (F_1,13_ = 5.58, *p* = 0.036; Figure 4). Although the raw data suggest that this effect was greatest in the aseptic regimen, there was no formal statistical support for this rearing × inoculation interaction (F_1,12_ = 0.014, *p* = 0.9). However, inspection of model fits indicated that there was more heterogeneity in the data for the standard rearing treatment and the low proportions in this treatment mean we have low power to resolve the symbiont inoculation treatment in standard rearing conditions. A conservative analysis shows a clear and strong effect of *Ec* inoculation using the data from the aseptic regime only (F_1,16_ = 5.1, *p* = 0.0033; Figure 4).

## 4. Discussion

We tested whether exerting greater control over the microbiome of insects in larval culture could improve their fitness as adults. Excluding potential microbial contaminants by rearing larvae in near-aseptic conditions substantially increased adult fitness; inoculating larvae with a known enteric symbiont provided further improvements in male pupal weight and competitive fitness. The standard larval rearing conditions for *P. xylostella* have had a long period of optimization, and it was beyond the scope of this study to fully optimize rearing conditions in our aseptic setup. Unsurprisingly, there was evidence that larval rearing conditions were not ideal in the aseptic setup: Pupal weights were lower and average survival of larvae marginally reduced (albeit non-significantly). Clearly, the aseptic rearing could be improved either by designing custom plasticware, or experimentally assessing the optimal egg load per container. Nevertheless, improvements in adult fitness occurred in the aseptic regime despite these sub-optimal larval conditions. Moreover, the results indicate that ‘probiotic’ manipulation of the larval microbiome had a positive effect on the fitness of adult DBM, over and above that of near-aseptic rearing conditions. These data are in contrast to previous studies with *Enterobacter* and transgenic medfly, in which gut bacteria enhanced larval survival but did not improve male mating fitness [37]. More broadly, our results suggest that the lepidopteran microbiome, despite its low diversity and the prevalence of many transient species [38], can have significant impacts on fitness.

In standard conditions, *E. cloacae* exhibited slight parasitism in terms of reduced pupal weight in the presence of the *E. cloacae*, in agreement with previous studies of this *E. cloacae* in DBM [39]. The overall results suggest a condition-dependent mutualism between *E. cloacae* and SL males. In insects, host-bacteria interactions are well documented [27,28] and more specifically, symbioses can flip from parasitic to commensal or mutualistic according to environmental conditions, such as diet quality [39,40,41]. In this study, diet was initially sterile and consisted of the same components in aseptic and standard conditions. Even in our near aseptic rearing conditions (Figure 1), the presence of non-culturable bacteria can still occur, although it is relatively straightforward to exclude culturable microbes [22,42]. While we did not characterize the non-culturable community in these experiments, variation in abundance or community composition is a plausible cause of the impact of rearing conditions on adult fitness. The additional benefit of *Ec* inoculation could therefore arise from additional benefits in reducing the presence of pathogens or parasites [43].

There are alternative explanations for the improved growth and fitness of *E. cloacae*-inoculated males. Gut bacteria can improve efficiency of digestion or provide essential nutrients [27]. There is also a link between gut bacteria and nutrient uptake, particularly with respect to nitrogen [29]. *Enterobacter* populations can contribute to the production of dinitrogen reductase in insects [44], an important enzyme involved in nitrogen fixation [45]. Enhancement of sexual signalling by symbionts also occurs in several species [31,46,47]. In locusts, the gut microbiota (containing *Enterobacter* species) increase pheromone production, which in turn increase aggregations and mating success [46]. In *P. xylostella*, males secrete chemical signals in the presence of females, from a specialised hair-pin gland found on their abdomens, which has a putative role in sexual signalling during courtship [48]. However, since the benefits of *E. cloacae* inoculation depend on rearing conditions in this study, nutritional or signalling explanations for these results are not the most parsimonious, since *E. cloacae* densities were indistinguishable in both standard and aseptic rearing conditions. Benefits of *Enterobacter* infection, for instance, would have to depend on the presence of other elements of the microbiota. Alternatively, since the removal of gut microbes can perturb insect metabolism [32], one explanation for the condition-dependent benefit of *E. cloacae* is that gut microbes normalize insect physiology and offset some of the potential side effects of rearing in near-aseptic conditions. Overall, further research is needed in order to depict the exact mechanisms underlying the benefits of the host-bacteria symbiosis investigated here.

## 5. Conclusions

While aseptic rearing methods and gut microbiota inoculation are not essential for the high fitness of transgenic insects [49], inclusion of these extra controls may make rearing methods more robust and less variable between laboratories [21]. For large-scale insect releases, rolling out robust rearing regimes across a number of sites is likely to be operationally important. The simple innovation of incorporating antibiotic into diet, and inoculating insects with symbiotic bacteria that are resistant to that antibiotic, provides a readily transferable tool for other insect-rearing systems. If incorporated into transgenic pest insects under the right abiotic conditions, gut bacteria could potentially contribute to the enhanced controlling of pest populations at lower costs, which in turn could contribute to the reduction of crop destruction or disease transmission [15,37].

## Figures and Tables

**Figure 1 insects-10-00089-f001:**
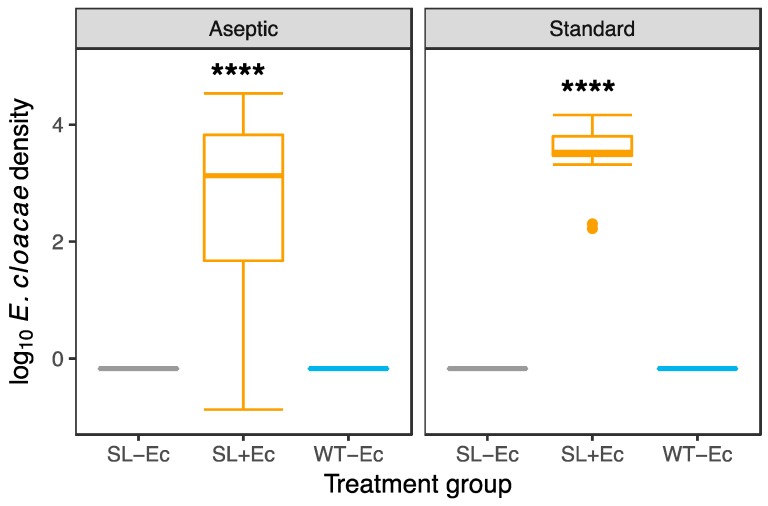
Culturable bacterial densities in inoculated (SL+Ec) and symbiont-free treatment groups (−Ec) under aseptic and standard conditions. Bacterial densities were assessed in self-limiting (SL) transgenic larva that had *Enterobacter cloacae* added to their diet (SL+Ec) and compared to SL larva without *E. cloacae* in their diet (SL−Ec) and wild type (WT) larva without *E. cloacae* in their diet (WT−Ec). In aseptic conditions, in the SL+Ec treatment, 94 out of 96 larvae were successfully inoculated with JJBC. In standard conditions, all SL+Ec (*n* = 8) contained the focal symbiont. In both standard and aseptic conditions, treatments that were not inoculated with *Ec* had bacterial densities of 0 c.f.u./μL (WT−Ec (aseptic: *n* = 36, standard *n* = 7) and SL−Ec (aseptic *n* = 60, standard *n* = 8). Bacteria in the inoculated larvae were streptomycin resistant and confirmed in morphology to our focal *E. cloacae* JJBC 11.1B strain. Asterisks indicate that inoculated treatments SL+Ec contain significantly more microbes than uninoculated treatments (*p* < 0.0001 in Kruskal-Wallis tests).

**Figure 2 insects-10-00089-f002:**
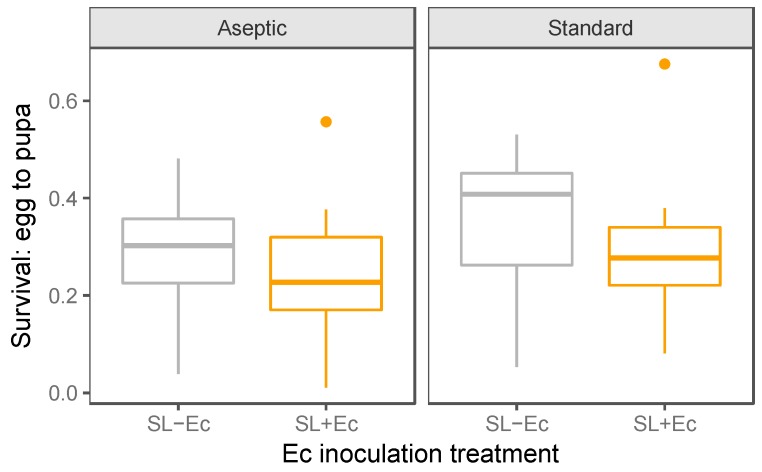
Survival egg to pupa for self-limiting insects in both aseptic and standard conditions, and with and without *E. cloacae* inoculation (+Ec–Ec). Eggs were counted on parafilm strips before being added to rearing containers, and the numbers of healthy pupae recorded after seven days. In the aseptic experiment, survival was recorded in 25–26 Petri dishes in each inoculation treatment, while under standard condition survival was assessed in 11–12 tubs per treatment.

**Figure 3 insects-10-00089-f003:**
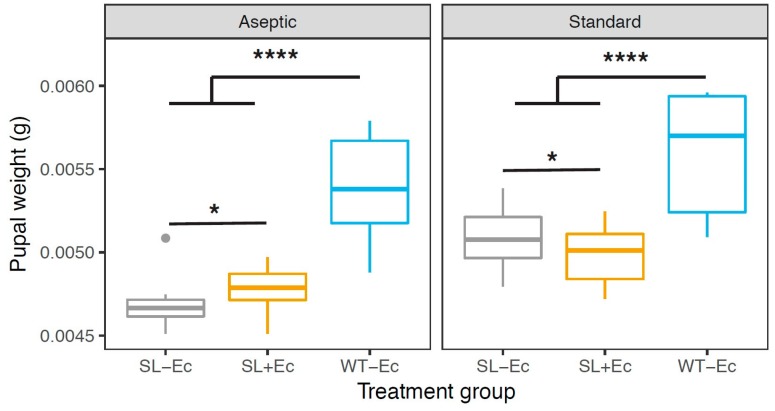
The effect of rearing regime, genotype, and inoculation with gut bacteria (*E. cloacae*) on the distribution of male pupal weights. ‘SL−Ec’ represents self-limiting transgenic *Plutella xylostella* pupae without *E. cloacae*, ‘SL+Ec’ represents self-limiting transgenic *P. xylostella* pupae with *E. cloacae*, and ‘WT−Ec’ represents wild type *P. xylostella* pupae without *E. cloacae*. For aseptic conditions, male pupae were weighed from 35 Petri dishes in each SL treatment group (700 pupae in all) and 12 Petri dishes in the WT−Ec treatment group (110 pupae). Standard rearing conditions used 11–12 replicate tubs (674 SL pupae and 85 WT pupae). Asterisks indicate treatment contrasts showing significant differences between treatment groups (* *p* <0.05, **** *p* < 0.0001).

**Figure 4 insects-10-00089-f004:**
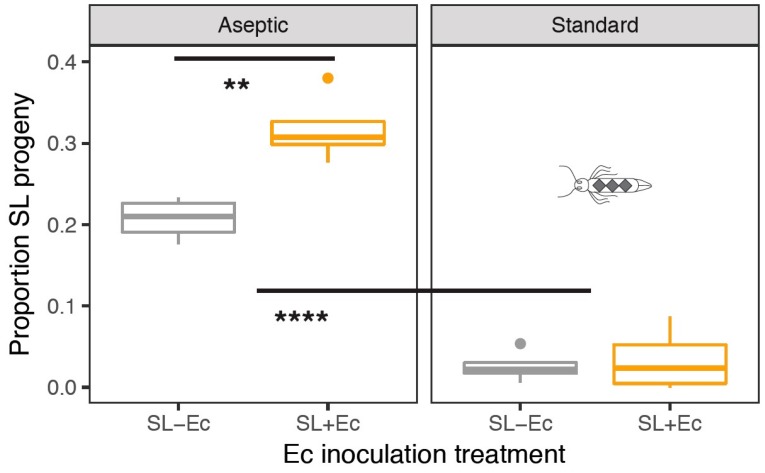
The impact of rearing regime and inoculation with *E. cloacae* gut bacteria on fitness of self-limiting males in mating competition experiments. Here, fitness is measured from the proportion of transgenic fluorescent male SL progeny in cages with 10:1 SL: Wild type release ratio and 120 insects per cage. In the aseptic regime, we scored a minimum of 100 progeny per mating cage; in the standard treatment, 200–400 progeny were scored per cage. Asterisks indicate significantly different treatment contrasts (** *p* < 0.01, **** *p* < 0.0001).

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
