# Peer review of "Aseptic Rearing and Infection with Gut Bacteria Improve the Fitness of Transgenic Diamondback Moth, Plutella xylostella"

_insects, 2019, doi:10.3390/insects10040089_

Round 1

Reviewer 1 Report

I would like to congratulate the authors on a well conceived and executed study, interest in the interplay between gut microbes and host health is topical and of general interest, this work adds to a growing body of studies, that over time will allow us to better understand these relationships

. I have several minor points that I believe could help improve the manuscript, but no major concerns. I list my two biggest points here first, then go through the rest of the MS with some line by line suggestions:

One stand out point - is the authors use of gnotobiotic. I and many others would expect this to refer to insects carrying either one or a few defined microbes. However it seems to refer here to the antibiotic cleared insects, which although not explicitly proven as such would be better referred to as axenic, while the add-back experiments produce gnotobiotic insects. 

I am also not entirely convinced here by the authors claims that they have successfully created germ free insects - as the methdod by which this is checked relies on bacteria being culturable. 

Are there any other studies detailing the Plutella xylostella microbiome - which might support this methodology?

Alternatively could the authors run a qPCR for 16s on their germ free moths - to complement their culture dependent technique?

 I have several minor points that I believe could help improve the manuscript, but no major concerns.

L69 -  Male reproductive behavioura in any SIT or genetic control strain may also be severely impaired relative to wildtype simply through the process of laboratory adaptation. Many studies on insect lab colonisation show the rapid adaptation that occurs within a handful of generations. Arguably this may have a greater impact than any subsequent transgenesis.  

The authors then describe that a critical aspect of this study is the previously observed variation in the diamondback moth between laboratories. But they do not elaborate on this point. I assume the implication is the potential impact of microbes? Rather than the effect of different genetic strains or population bottlenecking?

L83 - Briefly mentioned the potential impact of gut microbes on mating pheromones, and reference impacts in Drosophila. But there have been several contentious papers looking directly at the impact of specific microbes on mate choice here - might be worth considering a brief mention. I note the authors were pointed towards looking at microbes and mate competition. 

Sharon G, et al. (2010) Commensal bacteria play a role in mating preference of Drosophila

melanogaster. Proc Natl Acad Sci USA 107:20051-20056.

Njarro MA, Sumethasorn M, Lamoureux A, & Turner TL (2015) Choosing mates based on

the diet of your ancestors: Replication of non-genetic assortative mating in Drosophila

melanogaster. PeerJ 3:e1173.

Leftwich PT, Clarke NVE, Hutchings MI, & Chapman T (2017) Gut microbiomes and

reproductive isolation in Drosophila. Proc Natl Acad Sci U S A 114:12767-12772.

L97 - in reference to the self-limiting system, but perhaps a little more explanation. It is the use of the dsx intron sequence put into the CDS of tTAV to produce functional protein in females but not males. In its current sentence it reads almost as though dsx is the lethal gene. 

L101 - could the authors briefly describe the strain Vero Beach, or explicitly link to a reference?

L151 - Data analysis - the version of R used should be included, along with any supplemental packages used e.g. for post hoc comparisons. Also they mention later that model simplification was used - it would be better if this was briefly mentioned in this section. It is also mentioned here that chi square tests were used, but I cannot determine where these are in the results. 

L166 - It would be helpful if the authors put glm in front of test statistics - to explicity differentiate whole model from post hocs. 

L203 - Presumably residuals were ok on this glm despite the very different sample sizes?

L235 - but survival and weight both went down slightly? Do they mean the male reproductive fitness of the SL line only? 

L283 - Also see Harvey-Samuel 2015. 

Author Response

Referee  1.   Responses in italics.

I would like to congratulate the authors on a well conceived and executed study, interest in the interplay between gut microbes and host health is topical and of general interest, this work adds to a growing body of studies, that over time will allow us to better understand these relationships. I have several minor points that I believe could help improve the manuscript, but no major concerns. I list my two biggest points here first, then go through the rest of the MS with some line by line suggestions:

One stand out point - is the authors use of gnotobiotic. I and many others would expect this to refer to insects carrying either one or a few defined microbes. However it seems to refer here to the antibiotic cleared insects, which although not explicitly proven as such would be better referred to as axenic, while the add-back experiments produce gnotobiotic insects. 

I am also not entirely convinced here by the authors claims that they have successfully created germ free insects - as the methdod by which this is checked relies on bacteria being culturable. 

Our use of the term gnotobiotic was intended to be in the sense of “known microbiome” i.e. known to contain Enterobacter or not- and therefore to encompass axenic and Enterobacter infected insects.  However we can see that this might be open to confusion- we have therefore now described our procedure as ‘aseptic rearing’.   We have tried to avoid using the term axenic in all experiments, as we have not conclusively proven the absence of microbes, as this referee points out. 

Are there any other studies detailing the Plutella xylostella microbiome - which might support this methodology?

Alternatively could the authors run a qPCR for 16s on their germ free moths - to complement their culture dependent technique?

Response- we have validated this rearing method with 16S PCR previously (Raymond et al 2009 Env Microbiol)- it does produce germ-free insects although we have not explicitly confirmed this for experimental larvae  in this study.  Our aim at the time was not to pursue mechanisms but to see if cleaner rearing conditions that could exclude contaminants would improve fitness.

 I have several minor points that I believe could help improve the manuscript, but no major concerns.

L69 -  Male reproductive behavioura in any SIT or genetic control strain may also be severely impaired relative to wildtype simply through the process of laboratory adaptation. Many studies on insect lab colonisation show the rapid adaptation that occurs within a handful of generations. Arguably this may have a greater impact than any subsequent transgenesis. 

Yes a reasonable point we have now included a reference to a recent review on this topic  (Hoffman AA & Ross PA 2018 Journal of Economic Entomology 111;p501-509 https://doi.org/10.1093/jee/toy024.

The authors then describe that a critical aspect of this study is the previously observed variation in the diamondback moth between laboratories. But they do not elaborate on this point. I assume the implication is the potential impact of microbes? Rather than the effect of different genetic strains or population bottlenecking?

we have re-structured the introduction to make this clearer.

L83 - Briefly mentioned the potential impact of gut microbes on mating pheromones, and reference impacts in Drosophila. But there have been several contentious papers looking directly at the impact of specific microbes on mate choice here - might be worth considering a brief mention. I note the authors were pointed towards looking at microbes and mate competition. 

Sharon G, et al. (2010) Commensal bacteria play a role in mating preference of Drosophila

melanogaster. Proc Natl Acad Sci USA 107:20051-20056.

Njarro MA, Sumethasorn M, Lamoureux A, & Turner TL (2015) Choosing mates based on

the diet of your ancestors: Replication of non-genetic assortative mating in Drosophila

melanogaster. PeerJ 3:e1173.

Leftwich PT, Clarke NVE, Hutchings MI, & Chapman T (2017) Gut microbiomes and

reproductive isolation in Drosophila. Proc Natl Acad Sci U S A 114:12767-12772.

We have drawn attention to the conflicting literature on mate choice (Leftwich et 2017) although we do not believe that the older literature on bacteria producing precursors for pheromones is controversial.

L97 - in reference to the self-limiting system, but perhaps a little more explanation. It is the use of the dsx intron sequence put into the CDS of tTAV to produce functional protein in females but not males. In its current sentence it reads almost as though dsx is the lethal gene. 

Edited to emphasize that tTAV is lethal effector.

L101 - could the authors briefly describe the strain Vero Beach, or explicitly link to a reference?

not much is known about this population but we have referred to the Jin et al paper which describes it in a bit more details.

L151 - Data analysis - the version of R used should be included, along with any supplemental packages used e.g. for post hoc comparisons. Also they mention later that model simplification was used - it would be better if this was briefly mentioned in this section. It is also mentioned here that chi square tests were used, but I cannot determine where these are in the results. 

We have modified this section- the mention of Chi square tests has been deleted- this should have been edited out in an earlier revision.

L166 - It would be helpful if the authors put glm in front of test statistics - to explicity differentiate whole model from post hocs. 

OK.

L203 - Presumably residuals were ok on this glm despite the very different sample sizes?

Yes, fair point residuals not ideal because of the many zeroes, but there a difference in 3-4 orders of magnitude here between inoculated & un-inoculated insects so this is a very robust result.  We have replaced these tests with non-parametric analysis.

L235 - but survival and weight both went down slightly? Do they mean the male reproductive fitness of the SL line only? 

Yes – we measured SL male fitness- this has been clarified in the figure 4 legend.

L283 - Also see Harvey-Samuel 2015.

Reviewer 2 Report

It is well-known that the insect microbiome can have important effects on the fitness of their insect host. We are not talking about insect pathogens, but about microbes that provide essential nutrients, boost the immune system, change host behavior or affect sex determination. Here the authors study the fitness effects of gut microbes on the fitness of insects in cultivation, in particular of genetically modified insects that are being used in biocontrol programs. To establish the fitness effects of gut microbes is important to ensure that cultivated insects used in control strategies have the same fitness as their natural counterparts. This is particularly important in a GMO system that use antibiotics as a trigger as is the case in this study on Plutella moth that is based on a female-specific self-limiting gene. The authors conclude that the fitness of transgenic males of Plutella moth is increased by rearing them under sterile conditions and was further improved by adding one of their normal bacterial inhabitants Enterobacter claocae. This indicates that manipulation of the gut microbiome can be a useful method to improve fitness of cultivated insects.

I think you should incorporate two relevant papers to your study that look into the gut microbiome of Plutella in relation to herbivory and pesticide resistance: Xia et al. (2017) Metagenomic Sequencing of Diamondback Moth Gut Microbiome Unveils Key Holobiont Adaptations for Herbivory. Front. Microbiol., 26 April 2017 | https://doi.org/10.3389/fmicb.2017.00663 and Xia et al (2018). Gut Microbiota Mediate Insecticide Resistance in the Diamondback Moth, Plutella xylostella (L.).  Front. Microbiol., 23 January 2018 | https://doi.org/10.3389/fmicb.2018.00025. These papers also discuss the potential for exploiting the gut microbiome of Plutella in control programs. I am actually a little surprised that you did not mention them.

I think the material and methods can be improved”

Please clarify how Plutella eggs were treated under gnotobiotic rearings (line 116 and further). Under standard rearings you describe how Plutella eggs were treated line (line 114). Under gnotobiotic rearings Plutella treatment is not mentioned (line 116 and further).

It would have been nice to have included WT+Ec as a treatment group in addition to the three treatment groups SL-Ec, SL+Ec and WT-Ec. This would allow to assess whether Ec affects fitness in WT line as well. Please explain why you did not include this treatment group.

Line 133. Please describe the origin of the neonate larvae. Where they from the standard rearings-egg surface sterilization as mentioned in line 114?

Line 136: Why did you use different inoculation volumes in standard and gnotobiotic diets?

To avoid confusion: add that you used WT-Ec males in the competition experiments in line 142.

Please describe in more details in M&M what kind of positive and negative controls you used  to test your hypotheses. For example, how rearing uninfected larva on standard and gnotobiotic diet could inform you about the potential negative effects of antibiotics in the diet on Plutella fitness.

Line 201: Why would you want to simplify the model, especially since you  show that rearing method has different effects on pupal weight of the treatment groups.

Statistics:

I would like to see a bit more explanation on the various statistical test you do. For example in line  163 you state the results of a t-test based on the data in figure 1. But what kind of t-test did you do: paired and what pairs did you make? In line 178 and further you seem to perform a GLM test, please provide details of the model, what factors, what interactions did you investigate.

Author Response

Referee 2.  Responses in italics

It is well-known that the insect microbiome can have important effects on the fitness of their insect host. We are not talking about insect pathogens, but about microbes that provide essential nutrients, boost the immune system, change host behavior or affect sex determination. Here the authors study the fitness effects of gut microbes on the fitness of insects in cultivation, in particular of genetically modified insects that are being used in biocontrol programs. To establish the fitness effects of gut microbes is important to ensure that cultivated insects used in control strategies have the same fitness as their natural counterparts. This is particularly important in a GMO system that use antibiotics as a trigger as is the case in this study on Plutellamoth that is based on a female-specific self-limiting gene. The authors conclude that the fitness of transgenic males of Plutellamoth is increased by rearing them under sterile conditions and was further improved by adding one of their normal bacterial inhabitants Enterobacterclaocae. This indicates that manipulation of the gut microbiome can be a useful method to improve fitness of cultivated insects.

I think you should incorporate two relevant papers to your study that look into the gut microbiome of Plutellain relation to herbivory and pesticide resistance: Xia et al. (2017) Metagenomic Sequencing of Diamondback Moth Gut Microbiome Unveils Key Holobiont Adaptations for Herbivory.Front. Microbiol., 26 April 2017 | https://doi.org/10.3389/fmicb.2017.00663and Xia et al (2018). Gut Microbiota Mediate Insecticide Resistance in the Diamondback Moth, Plutella xylostella(L.).  Front. Microbiol., 23 January 2018 | https://doi.org/10.3389/fmicb.2018.00025. These papers also discuss the potential for exploiting the gut microbiome of Plutellain control programs. I am actually a little surprised that you did not mention them.

 You are right in that we should have cited the Xia et al 2017 paper.   The second paper does have less direct relevance for this MS, however.

I think the material and methods can be improved”

Please clarify how Plutellaeggs were treated under gnotobiotic rearings (line 116 and further). Under standard rearings you describe how Plutellaeggs were treated line (line 114). Under gnotobiotic rearings Plutellatreatment is not mentioned (line 116 and further).

We used the same surface sterilization protocol in both rearing treatments- this has now been clarified.

It would have been nice to have included WT+Ec as a treatment group in addition to the three treatment groups SL-Ec, SL+Ec and WT-Ec. This would allow to assess whether Ec affects fitness in WT line as well. Please explain why you did not include this treatment group.

We have investigated the impact of Ec on growth of WT insects in other studies (Matthews et al 2019- now cited in discussion).  Here our main objective wasto explore factors affect the fitness of the GMO insects. We had to produce WT insects for mating experiments so we recorded some life history traits but we did not conduct the full inoculation experiments.  Note that we could not assess effect of Ec inoculation on mating fitness for WT insects – to do this we needed the RFP marked insect line.

Line 133. Please describe the origin of the neonate larvae. Where they from the standard rearings-egg surface sterilization as mentioned in line 114?

We have clarified this sentence.

Line 136: Why did you use different inoculation volumes in standard and gnotobiotic diets?

The standard rearing tubs have a slightly wider diameter- this means slightly more fluid to cover food- we have clarified this sentence.

To avoid confusion: add that you used WT-Ec males in the competition experiments in line 142.

OK.

Please describe in more details in M&M what kind of positive and negative controls you used  to test your hypotheses. For example, how rearing uninfected larva on standard and gnotobiotic diet could inform you about the potential negative effects of antibiotics in the diet on Plutellafitness.

Our hypotheses was directional here- we predicted that increased control of contamination would increase fitness – this is largely what we saw.   If antibiotics had decreased insect fitness we would have seen lower fitness in the aseptic rearing methods. However we did not disentangle the effects of antibiotic from the whole aseptic rearing treatment.

Line 201: Why would you want to simplify the model, especially since you  show that rearing method has different effects on pupal weight of the treatment groups.

The model shows that there are treatment effects but not which treatment levels are different from one another.   The biggest effect here was the difference between WT and SL insects- the model simplification shows that there was also a significant different difference between SL insects with and without E. cloacae.

Statistics:

I would like to see a bit more explanation on the various statistical test you do. For example in line  163 you state the results of a t-test based on the data in figure 1. But what kind of t-test did you do: paired and what pairs did you make? In line 178 and further you seem to perform a GLM test, please provide details of the model, what factors, what interactions did you investigate.

 We have given more details of the statistical approaches in the methods section.  Following comments from referee 1 we decided to replace the analysis of bacterial counts with non-parametric tests.

In section 3.2 we have listed test for main effects and their interaction on survival & clarified where we are testing interactions on pupal weight.

In section 3.3 we have also listed tests for main effects and interactions. 

Reviewer 3 Report

In the paper "Gnotobiotic rearing and controlled infection with gut symbionts improve adult fitness in transgenic diamondback moth, Plutella xylostella" authors Sommerville, Zhou and Raymond compare the fitness traits of transgenic insects that have been reared in gnotobiotic conditions with or without the addition of a potential probiotic. While the work they present in this paper is not innovative, it brings important results that could be used to optimize the efficiency of transgenic insects for population control purpose. Their approach is simple but sound. I have only one major concern: the quality of the writing for the Abstract and Introduction could be greatly improved as these parts are very unpleasant to read (concepts are introduced in a very confusing way or they lack definition). If they improve the writing of these parts, I think this article will be suitable for publication in Insects. Following are some other minor comments that could be used to improve this article:

Title: the title is very complex to understand. What does "controlled infection with gut symbionts" is supposed to mean? 

Abstract: the abstract is very confusing to read. It alternates long sentences with numerous ideas, and short sentences that seems coming out of nowhere. I would suggest the authors to improve the writing of their abstract as it would improve the visibility of their work.

l.31: I don't understand this sentence.

l.42-44: reference needed

l.78: not "maybe be" just "may be"

l.87: for clarity purpose I think you should explain better how you define "gnotobiotic" in your system. Gnotobiotic usually refers to the condition of the organism, but it seems that here you refer only to rearing condition. Do you mean that you rear the insects under controlled/known microbial environment?

l.155: what graphical method did you use? A q-q plot? You should specify what method you used.

l.161: what about the nonculturable bacteria? You should maybe consider a more resolutive method like PCR with universal 16S primers.

l.163&166: you are presenting a "t" statistic, usually resulting from a Student t-test but you never wrote that you used such a test in the methods. I assume you meant "simple Student t-test" in your methods instead of "Chi-square"

l.181: if I understand correctly, you are presenting a F statistic from your GLM comparison (function anova(your_glm, test="F")). However, your model is based on the quasibinomial family, so the model comparison should be carried out using a chi-square test (function anova(your_glm, test="Chisq")

l.201: what post-hoc contrast did you use? Tukey? You should specify it.

l.218: same comment than earlier, it should be a Chi-square and not a F-test to do the model comparison.

Figures: please put the statistical results on your figures.

Author Response

Referee 3  Responses in italics

In the paper "Gnotobiotic rearing and controlled infection with gut symbionts improve adult fitness in transgenic diamondback moth, Plutella xylostella" authors Sommerville, Zhou and Raymond compare the fitness traits of transgenic insects that have been reared in gnotobiotic conditions with or without the addition of a potential probiotic. While the work they present in this paper is not innovative, it brings important results that could be used to optimize the efficiency of transgenic insects for population control purpose. Their approach is simple but sound. I have only one major concern: the quality of the writing for the Abstract and Introduction could be greatly improved as these parts are very unpleasant to read (concepts are introduced in a very confusing way or they lack definition). If they improve the writing of these parts, I think this article will be suitable for publication in Insects. Following are some other minor comments that could be used to improve this article:

Title: the title is very complex to understand. What does "controlled infection with gut symbionts" is supposed to mean? 

Abstract: the abstract is very confusing to read. It alternates long sentences with numerous ideas, and short sentences that seems coming out of nowhere. I would suggest the authors to improve the writing of their abstract as it would improve the visibility of their work.

We have simplified the language in the title and re-written the abstract in order to improve clarity.

l.31: I don't understand this sentence.

This has been re-written.

l.42-44: reference needed

 reference inserted

l.78: not "maybe be" just "may be"

corrected

l.87: for clarity purpose I think you should explain better how you define "gnotobiotic" in your system. Gnotobiotic usually refers to the condition of the organism, but it seems that here you refer only to rearing condition. Do you mean that you rear the insects under controlled/known microbial environment?

 In response to comments  from referee 1 and this comment we have avoided using this term – instead we have described this method as an “aseptic rearing” method.

l.155: what graphical method did you use? A q-q plot? You should specify what method you used.

 Ok this has been edited.

l.161: what about the nonculturable bacteria? You should maybe consider a more resolutive method like PCR with universal 16S primers.

  A point also raised by referee 1.  In brief we have validated this method previously with PCR and decided it was not essential for hypothesis testing in this study.

l.163&166: you are presenting a "t" statistic, usually resulting from a Student t-test but you never wrote that you used such a test in the methods. I assume you meant "simple Student t-test" in your methods instead of "Chi-square"

This has been corrected.

l.181: if I understand correctly, you are presenting a F statistic from your GLM comparison (function anova(your_glm, test="F")). However, your model is based on the quasibinomial family, so the model comparison should be carried out using a chi-square test (function anova(your_glm, test="Chisq")

For well behaved binomial models we would report a chi-sq.  However, F tests are the correct test for quasibinomial models with overdispersed data.  See Crawley – R book or Venables & Ripley MASS.

l.201: what post-hoc contrast did you use? Tukey? You should specify it.

Now explained in M&M.

l.218: same comment than earlier, it should be a Chi-square and not a F-test to do the model comparison.

see comment above.

Figures: please put the statistical results on your figures.

We have summarized the results of treatment contrasts & post hoc tests on figures, where significant.

Insects EISSN 2075-4450 Published by MDPI AG, Basel, Switzerland RSS E-Mail Table of Contents Alert
Back to Top